# Advances in Electromagnetic Piezoelectric Acoustic Sensor Technology for Biosensor-Based Detection

Gábor Mészáros [1], Sanaz Akbarzadeh [2], Brian De La Franier [3], Zsófia Keresztes [1] and Michael Thompson [3,*]

1   Institute of Materials and Environmental Chemistry, Research Centre for Natural Sciences, H-1117 Budapest, Hungary; meszaros.gabor@ttk.hu (G.M.); keresztes.zsofia@ttk.hu (Z.K.)
2   Department of Chemistry, Yasouj University, Yasouj 75918-74831, Iran; akbarzade.sanaz@yahoo.com
3   Lash Miller Laboratories, Department of Chemistry, University of Toronto, 80 St. George Street, Toronto, ON M5S 3H6, Canada; brian.delafranier@mail.utoronto.ca
*   Correspondence: m.thompson@utoronto.ca; Tel.: +1-416-978-3575

**Abstract:** The ultra-high frequency EMPAS (electromagnetic piezoelectric acoustic sensor) device is composed of an electrode-less quartz disc in which shear oscillation is induced by an AC-powered magnetic coil located 30 μm below the substrate. This configuration allows the instigation of high acoustic harmonics (in the region of 49th–53rd), with the resulting enhanced analytical sensitivity for biosensor purposes compared to the conventional thickness-shear mode device. In this paper, we introduce significant improvements to the operation of the system with respect to sensing applications. This includes a new interface program and the capability to measure the acoustic quality factor not available in the prototype version. The enhanced configuration is subject to testing through biosensor detection of surface adsorption of biological macromolecules, which include β-casein, and a gelsolin-actin complex.

**Keywords:** acoustic wave detection; electromagnetic piezoelectric acoustic sensor; biological macromolecule detection

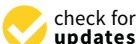

## 1. Introduction

Biosensor technology continues to attract significant interest with respect to the assay of biochemical species, especially in the medical care field [1–3]. In this regard, such measurement by various types of acoustic wave-based sensor has figured prominently [4]. The reasons for this lie in the capabilities of acoustic wave devices to produce sensitive, real-time and label free detection of biological entities of interest not only with respect to bioanalytical applications, but also from a fundamental biophysical perspective. Applications encompass liquid phase studies of surface biochemical macromolecule adsorption, immunochemical, enzyme and nucleic acid-based detection, and study of the behavior of cells and various particles at interfaces. A number of different devices have been employed in order to instigate acoustic wave signaling of such events. These include the well-known thickness-shear mode sensor (TSM), often referred to as the quartz crystal microbalance [5], and devices based on shear horizontal surface acoustic waves [6] and love wave propagation [7].

An additional sensor configuration introduced in the early 2000s by our group in collaboration with the University of Cambridge, UK is that based on ultra-high frequency acoustic wave technology, the electromagnetic acoustic sensor (EMPAS) [8–10]. Essentially, the sensor is a hybrid composed of the conventional TSM with a quartz crystal, and the magnetic acoustic resonator sensor (MARS) introduced by Stevenson et al. [11] where acoustic waves are instigated in wafers of glass. In summary, the EMPAS system involves the use of an oscillating electromagnetic field from an alternating current (AC) powered electromagnetic coil to remotely excite an ultra-thin piezoelectric substrate such as quartz, gallium phosphate [12] or aluminum nitride [13] at its acoustic resonance. The coil is

placed directly below the crystal at a specified distance of separation (Figure 1). In order to conduct experiments the device is incorporated into a flow-cell for injection of reagents.

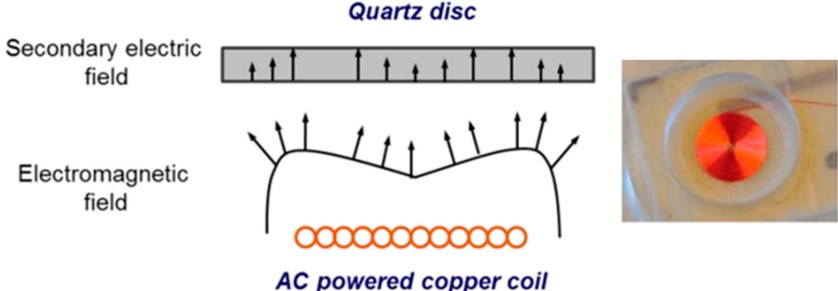

**Figure 1.** Excitation of quartz disc via the secondary electric field from an AC-powered copper coil (shown on the right).

The EMPAS system has been applied to a wide variety of bioanalytical and biophysical research problems, where a particular emphasis has been placed on the study of non-specific adsorption or fouling at silica surfaces. The first example of this undertaking was the investigation of functionalizable mixed self-assembled monolayers (SAMs) produced from linker and diluent molecules [14]. Biotinthiol could be subsequently immobilized for detection purposes in a straightforward and coupling-free manner, and specific and non-specific adsorptions of avidin was measured at a frequency of 0.82 GHz. In a further study, it was demonstrated that attachment of immobilized oriented antibody Fab' fragments to linkers reduced the fouling levels of surfaces from albumin and serum by up to 80% compared to the surfaces without fragments [15]. This phenomenon was attributed to the possibility that antibody fragments increase the hydration of the surfaces leading to the formation of an anti-fouling water barrier. In a similar vein, the ability of polymer brushes grafted onto the quartz-based EMPAS device via an adlayer was assessed in terms of the significant prevention of protein adsorption from blood plasma [16]. The results were compared with experiments involving blood in interaction with brushes on polycarbonate substrates. Finally, with regard to EMPAS study of fouling from proteins, adsorption on quartz was investigated recently with respect to varying lengths of self-assembled monolayers on the sensor surface [17].

In addition to the fouling research outlined above, several bioanalytical-oriented studies have been conducted with the EMPAS device. The first such application was the use of the sensor to detect HIV antibodies in serum [18]. An HIV-2 probe was attached to the quartz surface in order to examine interaction with anti-gp36 and anti-gp41 monoclonal antibodies in a solution containing serum. The system was operated at 1.06 GHz. A second study dealt with detection of cocaine via binding to linker surface-attached MN4 aptamer [19]. The EMPAS device detected the binding of the drug through target mass loading coupled to aptamer tertiary structure folding and achieved an apparent Kd of 45,712 mM, and a limit of detection of 0.9 mM. Shortly after this work the sensor was employed to the breast and prostate cancer metastasis biomarker, parathyroid hormone-related peptide (PTHrP) through binding to sensor-immobilized whole anti-PTHrP antibodies and Fab' fragments as biorecognition elements [20]. The whole antibody-based mass-amplified biosensor yielded the lowest limit of detection (61 ng/mL), highest sensitivity, and a linear range from 61 ng/mL to 100 μg/mL. The activity of plasmin, a component of milk, was detected through a sensor-attached casein probe [21]. The sensitivity of the EMPAS system produced measurements as low as 32 pM concentration of plasmin, reaching (and often exceeding) levels comparable to conventional techniques such as enzyme-linked immunosorbent assays (ELISA).

Despite the evident attractive features of EMPAS-based detection a number of technical issues remain to be solved. These include, for example, lack of shielding of components, efficient circuitry, capability for automation, and portability [22]. A number of significant

improvements are introduced in the present paper in terms of the equipment employed to record acoustic resonance parameters. Secondly, the improved configuration is tested through measurement of responses to non-specific surface adsorption of β-casein, a major protein component of milk, [23] and molecular reagents employed for the detection of ovarian cancer. The latter are the proteins, actin and gelsolin which are involved in the assay of lysophosphatidic acid (LPA), a biomarker for this form of cancer [24–26]. Surface-attached gelsolin is a probe for LPA and a method of assay has been developed based on replacement of fluorescent-labelled actin from the probe [27].

## 2. Materials and Methods

### 2.1. Materials and Components

Unless otherwise specified, all reagents including bovine β-casein were purchased from Sigma Aldrich, Oakville, Ontario. Gelsolin protein was produced by expression from PSY5 plasmids containing the gelsolin gene with a histidine tag as described previously [27]. The plasmids were kindly provided by Professor Robert Robinson of the University of Singapore. Protein mass and purity were determined by SDS-PAGE (12% acrylamide) and concentration by absorbance at 280 nm. Actin from rabbit muscle was purchased from Alfa Aeser or Sigma Aldrich and modified as outlined previously [27]. Protein concentration was determined by absorbance at 280 nM. Quartz crystals (AT-cut, 13.5 mm in diameter, 20 MHz fundamental frequency) were purchased from Lap-Tech Inc., Bowmanville, Ontario. The EMPAS coil circuit included a hand-wound flat spiral coil ($\approx$5 mm diameter) of polyurethane-coated copper wire (total diameter of metal and insulating layer, 105 $\mu$m). This coil facilitated the actual output of the electromagnetic field and received the feedback signal (Figure 1).

The EMPAS setup consisted of an HP 8648B (Agilent Technologies) radio frequency signal generator combined either with a SR510 digital lock-in amplifier (Stanford Research Systems) or a USB-6211 (National Instruments) data acquisition unit. Both setups also contained a laboratory-made, small analog circuit described previously [9]. The control program of the original setup was written in LabVIEW (National Instruments) while that of the modified version was coded in the GNU Dev-C++ environment in C.

### 2.2. Device Surface Modification

Quartz crystals were first sonicated in 20 mL of soap for 30 min. The crystals were successively rinsed with hot water followed by distilled water, and then gently dried with forced air followed by soaking in 6 mL of piranha solution pre-heated to 90 °C for 45 min using a hot water bath. The crystals were rinsed three times with distilled water, and three times with spectrograde methanol. Next, the substrates were sonicated in methanol for 2 min before being individually stored in vials and placed in an oven maintained at 150 °C for drying. After 2 h, the crystals were transferred into a 70%-maintained ($MgNO_3 \cdot 6H_2O$) humidity chamber for 24 h.

MEG-Cl [27] was diluted with anhydrous toluene (5 mL) under inert ($N_2$) and anhydrous ($P_2O_5$) atmosphere in a glovebox (1 $\mu$L/mL). The solution was added to glass vials (pre-silanized with trichloro(octadecyl)silane) containing quartz discs. The vials were capped and sealed with ParafilmTM M, removed from the glovebox, and placed on a spinning plate for 1.5 h. The discs were then rinsed with toluene and sonicated in toluene for 5 min followed by sonication in deionized water for 3 min. MEG-Cl discs were also modified by Ni-NTA by exposing them to a solution of ab-NTA and nickel (II) chloride (2 mg/mL ab-NTA, and 2 mg/mL nickel (II) chloride in deionized water, 4 mL). Pyridine (2 mL) was added before placement of the devices on a spinning plate overnight.

### 2.3. EMPAS Flow-through Measurements

The flow-through EMPAS measurements were generally conducted as in previous work on non-specific adsorption on quartz from protein solutions [17]. AT-cut 20.0 MHz quartz crystals, bare or surface modified, were placed in a flow-through system. One

side of the crystal was exposed to liquid, and the other side was exposed to air. Runs were performed with the crystal in the horizontal position and at ambient temperature. The crystal was secured in the holder using an O-ring. Each disc was thoroughly flushed with phosphate buffered saline (PBS) at a rate of 50 μL/min using a syringe pump. After ensuring uniform coverage of the disc with PBS (i.e., no bubbles), the resonant frequency of the device and quality factor were recorded as discussed above. After initializing the run, the signal was allowed to stabilize for 10–15 min. Then, a 50 μL portion of sample (protein solutions of either gelsolin 1-3 (0.3 mg/mL), actin (0.6 mg/mL), or bovine β-casein (0.05 mg/mL), in PBS buffer) was injected into the flow-through system using a low-pressure chromatography valve. This was followed with the uninterrupted flow of PBS at a rate of 50 μL/min to remove any non-specifically or loosely bound species. PBS was flowed until a stable baseline was achieved (10–15 min). Changes to the resonant frequency were noted during the course of the run. Prior to the introduction of a new disc, the system was vacuumed dry using a peristaltic pump and carefully dried.

## 3. Results and Discussion

### 3.1. EMPAS Architecture and Detection of Resonant Frequency

The original EMPAS setup consists of an HP 8648B high frequency signal generator and a Stanford Research Systems SR510 digital lock-in amplifier controlled by a Labview-based user interface. The output frequency of the HP 8648B slowly scans a narrow interval in the vicinity of the selected resonance frequency while it is modulated in frequency at the same time. The modulating audio frequency sinusoidal signal is provided by the lock-in amplifier. This modulated high frequency signal is fed to a planar coil supporting the quartz crystal. Since the impedance of the coil with the quartz depends on frequency, the frequency modulation will induce amplitude modulation the amplitude of which will be more or less proportional to the derivative of the impedance vs. frequency curve of the resonating element. The amplitude modulated signal can be easily demodulated by a simple diode rectifier. The in-phase component of the obtained audio frequency response signal is measured by the lock-in amplifier. This way instead of measuring the absolute value of the impedance of the resonating element, $|Z(f)|$, the resulting function is more or less identical to its derivative vs. frequency, $d|Z(f)|/df$.

The advantage of the method is obvious. First, the high frequency features of the resonance curve are transformed to the audio frequency range making the further handling of the signal much easier; second, the resulted signal vs. frequency curve will show (at least theoretically) zero crossing at the resonant frequency which is more reliable and easier to detect in comparison with a Lorentzian of Gaussian fitting of a peak. However, in reality the measured signal never shows a definite zero crossing since it is always superimposed on a slowly changing interfering signal due to the impedance of the planar coil. In addition, no information is gained on the dissipation factor corresponding to the selected resonance.

### 3.2. EMPAS Advances

To overcome the difficulties outlined above a new user interface program was prepared in C and the lock-in amplifier was exchanged to a 16-bit National Instruments data acquisition card (NI USB-6211) permitting the detection of the higher harmonic components of the audio frequency response signal as well (Figure 2). Since the interfering signal due to the impedance of the planar coil changes rather slowly with frequency the higher harmonics contain less and less contribution of it. The even harmonic components show a peak while the odd harmonic components show zero crossing at the resonant frequency. The third harmonic component practically does not contain any interference thus making it ideal for the detection of the resonant frequency.

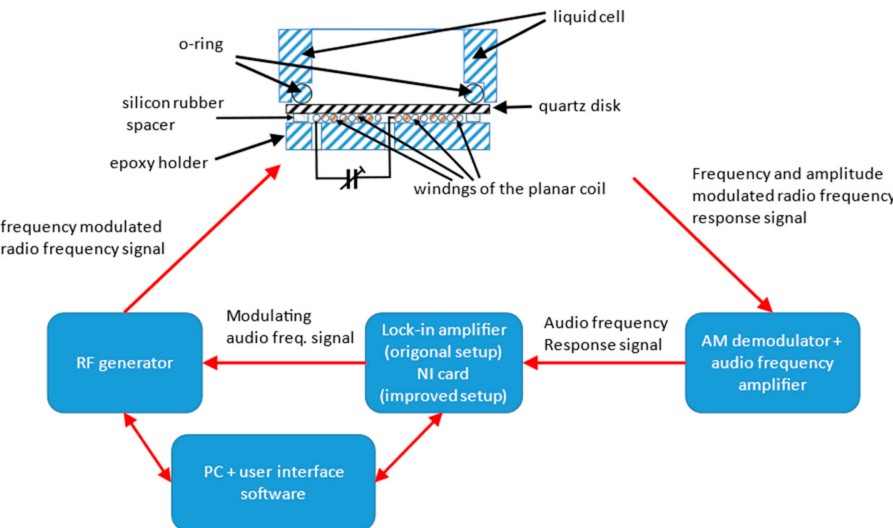

**Figure 2.** Schematic of improved electromagnetic piezoelectric acoustic sensor (EMPAS) configuration.

Here it should be noted that a resonance peak is slightly asymmetric to the center frequency. As a consequence, the peak position of the even harmonics and the zero-crossing position of the odd ones will show a small deviation from the resonant frequency. However, this small offset can be neglected since in the EMPAS technique, not the absolute value but the change of the resonant frequency is tracked.

The peak values of the even harmonics allow the calculation of the quality factor ($Q$) of the selected resonance. Namely, the ratio of the peak values of the 2nd and 4th harmonic components at the resonant frequency depend only on the $Q$ factor and the ratio of the resonant frequency and the deviation of the modulating audio frequency—this latter ratio is constant in an experiment. Thus, a simple one-dimensional iteration—which is involved in the new user interface program - permits the exact calculation of the value of the $Q$ factor. As for a demonstration, Figure 3 shows a test experiment on the adsorption of β-casein. In the numerical calculation the normalized transfer function of an resistor inductor capacitor (RLC) resonant circuit was applied as the mathematical model of the resonance:

$$|H(f)| = \frac{1}{Q\sqrt{\left(\frac{f}{f_0} - \frac{f_0}{f}\right)^2 + \frac{1}{Q^2}}} \tag{1}$$

where $f$ is the frequency, $f_0$ the resonant frequency and $Q$ the quality factor. The width of the peak referred to the $1/\sqrt{2}$ value is given by:

$$\Delta f = \frac{f_0}{Q} \tag{2}$$

With regard to a choice of modulation parameters, the time function of the output signal of the high frequency generator can be given as:

$$U(t) = U_0 \, \sin((f_{HF} + f_{dev} \, \sin(f_{AF} \, t) \, )t) \tag{3}$$

where $f_{\mathrm{HF}}$ is the high frequency, $f_{\mathrm{AF}}$ the modulating audio frequency and $f_{\mathrm{dev}}$ stands for the deviation of the frequency modulation with respect to $f_{\mathrm{HF}}$. The frequency of the modulation signal ($f_{\mathrm{AF}}$) should be chosen in such a way that the frequency of the measured higher harmonics (2nd, 3rd and 4th) keeps sufficient distance from the harmonic components of the mains. Convenient values are $f_{\mathrm{AF}} = nf_{\mathrm{MAINS}} \pm 10$ Hz (n = 1, 2, 3, . . . ). Good choices are the values 290 Hz or 310 Hz since those values fit in counties both with 50 Hz and

60 Hz mains frequency. The corresponding record length should be the integer multiple of 0.1 s but not less 0.2 s.

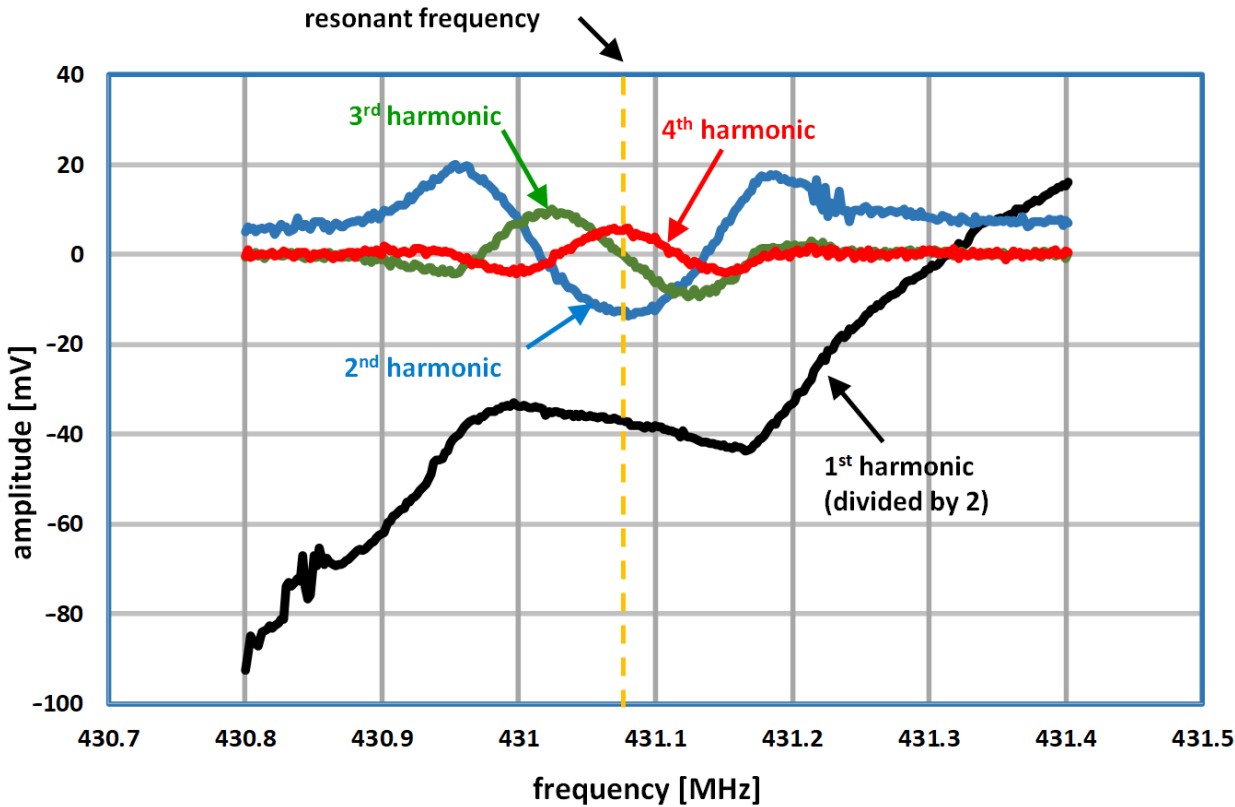

**Figure 3.** The amplitudes of the in-phase harmonic components of the demodulated audio frequency response are shown vs. frequency at a selected resonance frequency.

The deviation ($f_{dev}$) of the frequency modulation should be chosen in such a way that the demodulated response of the system contains sufficient higher harmonic content. On the other hand, too high of a $f_{dev}$ value will lead again to the disappearance of the resonance effect. A good choice can be the half of the width of the resonance peak, $f_{dev} = \Delta f / 2$ (see Equation (2)).

Measuring the harmonic components of the signal (Figure 3) found the modulating signal ($f_{AF}$) was 410 Hz, the deviation of the modulation ($f_{dev}$) was 125 kHz. The high frequency signal was stepped with 2 kHz between measurement points. Each point was calculated from an 800 ms long record. The curve of the 1st harmonic contains considerable interference; however, the supposed inflexion point at the resonant frequency is visible. In contrast, the curves corresponding to the higher harmonic components show clear features. The zero crossing of the 3rd harmonic allows the detection of the resonant frequency while the peak values of the 2nd and 4th harmonic components permits the calculation of the quality factor *Q*.

### 3.3. Test EMPAS Signals from Surface Interaction of Biological Macromolecules

In previous research, it has been shown that the EMPAS system can be employed for the detection of the enzyme plasmin in milk at low concentrations via interaction with device-adsorbed β-casein [21]. A distinct difference in enzyme activity was found for the protein immobilized on either hydrophilic or hydrophobic surfaces. In this work, a comparable signal of 22 KHz, at a resonant frequency of close to 1 GHz, was found for adsorption of the protein to the bare hydrophilic surface of the sensor (Figure 4). It has been suggested that such adsorption to a highly hydrophilic surface involves a conformational change caused by interaction of polar groups with hydroxyl functionalities

on the substrate [21]. Unlike the conventional EMPAS experiment, the improved system also yields values for the acoustic quality factor. The latter is equivalent to the widely recognized dissipation factor observed for a conventional TSM device. In this case, the Q value decreases following β-casein injection, precisely as would be expected for a dissipation factor. This indicates that surface adsorption results in an increased viscosity at the interface between the sensor and the solution.

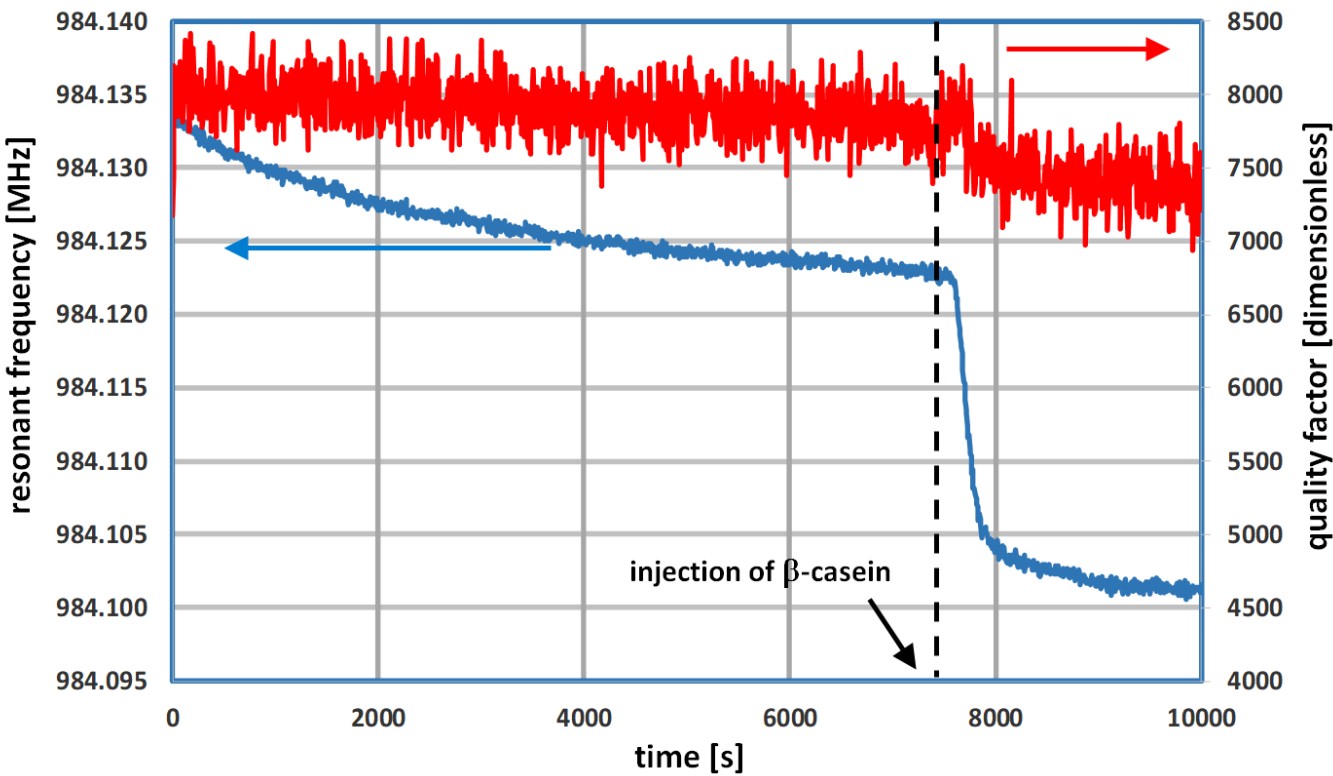

**Figure 4.** Test experiment involving surface adsorption of β-casein to hydrophilic quartz.

An initial experiment to evaluate the new EMPAS measurement system was to see if the protein gelsolin (a probe for LPA) could be bound selectively to the surface, and potentially extended with actin. To this end, PBS solution containing gelsolin was injected onto MEG-Cl coated crystals (should not bind), and MEG-Cl-Ni-NTA coated crystals (should bind to histidine tag of gelsolin) (Figure 5). This injection was followed by the injection of PBS containing actin protein to see if the actin would further bind to the gelsolin. In the case of MEG-Cl coated crystals where no binding was expected an initial dip in frequency is first observed for both the gelsolin and actin injections, followed by the signal returning to baseline (Figure 5 blue trace). This suggests there is some initial fouling of the MEG-Cl surface by the proteins, but this fouling washes off with continued PBS flow.

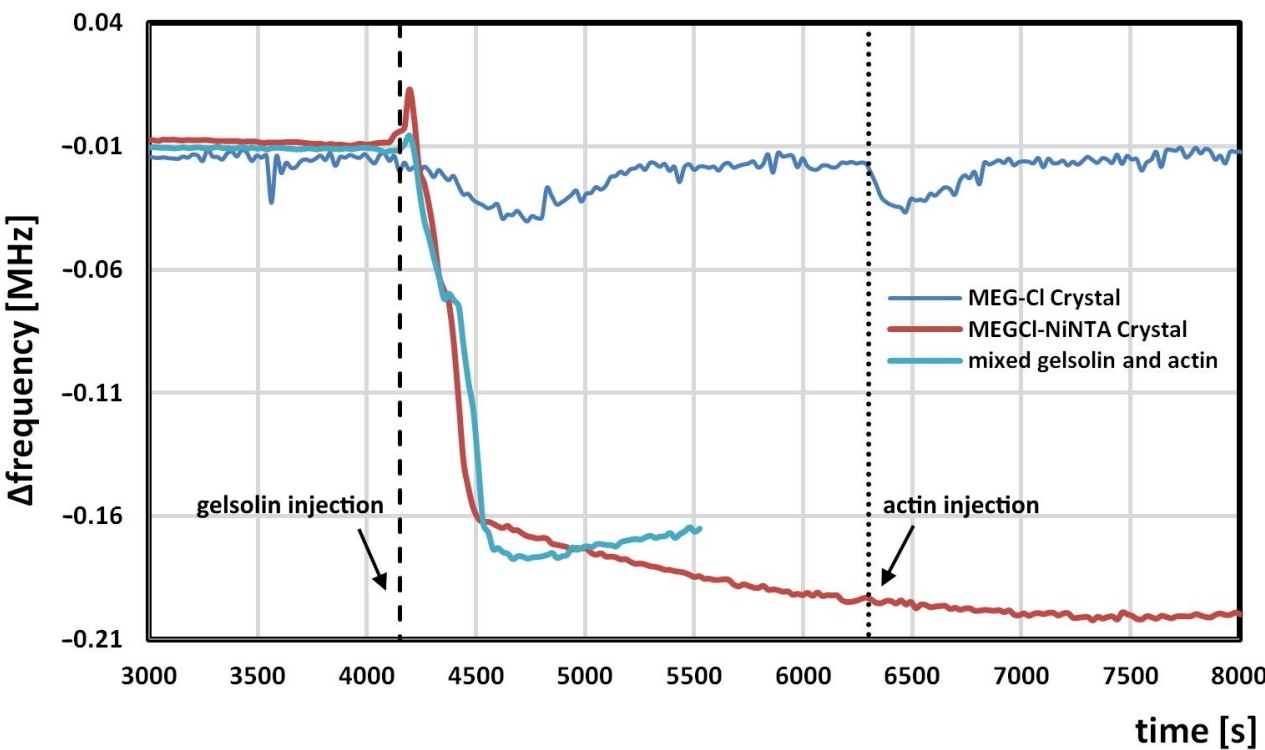

**Figure 5.** Initial EMPAS responses to gelsolin followed by actin injection on MEG-Cl and MEG-Cl-Ni-NTA coated crystals (blue and orange traces). Additionally, a mixed solution of gelsolin and actin was passed over MEG-Cl-Ni-NTA coated crystals (cyan trace).

On MEG-Cl coated crystals that have been extended with Ni-NTA there is a much larger frequency change observed following injection of gelsolin that remains through further PBS flow (Figure 5 orange trace). This suggests that gelsolin has been specifically bound to the surface via the Ni-NTA functional group. There was no observed change following injection of actin suggesting that actin was unable to bind to the surface bound gelsolin. This suggests that the gelsolin has packed the surface, and that actin is unable to reach the binding sites on gelsolin. To test this, a mixture of gelsolin and actin that were already bound together was injected over MEG-Cl-Ni-NTA coated crystals (Figure 5 cyan trace). A similar change in frequency was observed compared to when gelsolin alone was passed over the Ni-NTA containing surface, suggesting that surface saturation was achieved under both conditions. This result implies that in order to have the desired gelsolin-actin dual probe system for detecting the presence of LPA the proteins must be mixed together before surface deposition to ensure that the binding sites are not sterically hindered.

## 4. Final Remarks

The EMPAS sensor represents a strong contributor to the possible detection of biochemical moieties by acoustic wave propagation. This is associated with fact that the device can be operated at frequencies on, or close to, 1 GHz which enables highly sensitive detection of events instigated at the sensor surface. Furthermore, the employment of silica as an underlying interface offers a convenient surface for the attachment of chemical linkers and probes since there is very wide collection of chemistries available for such a purpose. However, a number of desirable improvements are warranted in order to enhance the overall efficiency and user friendliness of the system. In the present work, we have introduced changes to the programming and electronic side of the configuration, which clearly enhance the capability to recognize species at the quartz surface as evidenced by detection of biomolecular absorption. The data obtained with regard to the surface silanization and probe attachment was obtained with significant facility compared to the old set-up, thus

confirming that the new system functions remarkably well. The software and frequency driver used for the original system was difficult to use, and provided limited data with greater noise due to the limited ability to select harmonic frequencies.

In the future, the more physical aspects of the equipment require improvement. These include a more efficient method for the fabrication of the spiral coil, which is currently produced by hand, adjustment to allow ease of sensor introduction in to the flow cell, and possible conversion of the flow through system in to a microfluidic arrangement. Such efforts are underway.

**Author Contributions:** Conceptualization, M.T., Z.K. and G.M.; methodology, G.M., S.A. and B.D.L.F.; writing—M.T., B.D.L.F. and G.M.; review and editing—Z.K., G.M., and B.D.L.F.; All authors have read and agreed to the published version of the manuscript.

**Funding:** This work was funded by the European Union Horizon 2020 Research and Innovation Programme under the Marie Sklodowska-Curie grant agreement No 690898, the Development and Innovation Office of Hungary (NKFIH) BIONANO-GINOP 2.3.2-15-2016-00017, and by the Natural Sciences and Engineering Research Council of Canada (grant to M.T.).

**Acknowledgments:** We are very grateful to Robert Robinson of the University of Singapore for the provision of plasmids for the production of gelsolin. Yasouj University, Yasouj, Iran is gratefully acknowledged for research leave and support awarded to S.A.

**Conflicts of Interest:** The authors declare no conflict of interest.

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
