# Peer review of "Advances in Electromagnetic Piezoelectric Acoustic Sensor Technology for Biosensor-Based Detection"

_chemosensors, doi:10.3390/chemosensors9030058_

Round 1
Reviewer 1 Report
The paper describes the results obtained using a modified EMPAS system. Modification appear to consist mainly of a change in the signal detection method from a lock in amplifier to a high-resolution data acquisition card and the concomitant software changes. The use of the new detection method also allows a measure of the Q factor of the resonance. A number of examples of the application of this modified instrument is presented, showing the ability of the new system in the detection of different biochemical species.
Good results are shown for the operation of this sensor system, but no clear attempt is made to compare the performance of the improved system to the previous version. For example does the elimination of the lock in amplifier lead to any increased noise in the signal ? If so, how does this effect the sensitivity of the system to a target analyte ? A measurement of the Q factor during analyte adsorption is shown in Fig 3, but it not clearly explained whether any addition information is actually obtained from the Q value.
The “Final Remarks” section presented a good opportunity to discuss the merits of these system changes, but no attempt is made summarise it at this point. This results in a paper in which the goal is somewhat unclear: is it meant to present the enhanced capability of the sensor system or does it present a few random results that could have equally well been obtained by the previous instrument system ?
In summary: A well written paper that presents some interesting results particularly to the acoustic wave sensing community, but it can be improved by providing some more specific details and discussion on the changes in system performance and why this results in an improved EMPAS system.
The use of Fig. 1 also leads to some uncertainty. It appears to be the “standard” EMPAS system that the authors have use in previous work, but it is published with permission of Powertec, Rep. of Slovakia. Is it a commercial system that was used in previous work ? Would it be better to actually reference the source of this system ?
Reviewer 2 Report
Dear Authors,
Your work is devoted to a topical topic related to the study of the electromagnetic piezoelectric acoustic sensor. The presented results will undoubtedly be of interest to the international scientific community. At the same time, from the point of view of the reviewer, some additions should be made to the article. This is due to the interest that the reader can show in it. In particular, the data related to "a planar coil" (line 161 and further on in the article) are of interest):
- What is a planar coil? What is its size? What, at least in the most general form, is the method of making this coil?
- It is worth giving a sketch showing the reader the relative position of the elements described in the sensor (planar coil, quartz crystal, etc.), i.e. a schematic representation of the sensor design. Their size is also of interest.
It is also worth introducing some clarifications in the process of preparing sensor elements and samples. For example, in line 116 "Quartz crystals were first sonicated in 20 mL of soap for 30 min." What are the crystal processing modes? Or the type of ultrasound unit and its power?
